# Genetic Diversity of Genes Controlling Unilateral Incompatibility in Japanese Cultivars of Chinese Cabbage

**DOI:** 10.3390/plants10112467

**Published:** 2021-11-15

**Authors:** Yoshinobu Takada, Atsuki Mihara, Yuhui He, Haolin Xie, Yusuke Ozaki, Hikari Nishida, Seongmin Hong, Yong-Pyo Lim, Seiji Takayama, Go Suzuki, Masao Watanabe

**Affiliations:** 1Graduate School of Life Sciences, Tohoku University, Sendai 980-8577, Japan; 2Division of Natural Science, Osaka Kyoiku University, Kashiwara 582-8582, Japan; am.osaka.kyoiku@gmail.com (A.M.); ukiko0620@gmail.com (Y.H.); xiehaolin20@gmail.com (H.X.); yu.oz.7991@gmail.com (Y.O.); hkr.disney.12121998@gmail.com (H.N.); 3Department of Horticulture, College of Agriculture and Life Science, Chungnam National University, Daejeon 34134, Korea; sungmin201@cnu.ac.kr (S.H.); yplim@cnu.ac.kr (Y.-P.L.); 4Department of Applied Biological Chemistry, Graduate School of Agricultural and Life Sciences, The University of Tokyo, Tokyo 113-8657, Japan; a-taka@mail.ecc.u-tokyo.ac.jp

**Keywords:** allelic diversity, *Brassica rapa*, Chinese cabbage, dominant negative effect, gene duplication, pollen-stigma interaction, self-incompatibility, unilateral incompatibility

## Abstract

In recent years, unilateral incompatibility (UI), which is an incompatibility system for recognizing and rejecting foreign pollen that operates in one direction, has been shown to be closely related to self-incompatibility (SI) in *Brassica rapa*. The stigma- and pollen-side recognition factors (*SUI1* and *PUI1*, respectively) of this UI are similar to those of SI (stigma-side *SRK* and pollen-side *SP11*), indicating that *SUI1* and *PUI1* interact with each other and cause pollen-pistil incompatibility only when a specific genotype is pollinated. To clarify the genetic diversity of *SUI1* and *PUI1* in Japanese *B. rapa*, here we investigated the UI phenotype and the *SUI1*/*PUI1* sequences in Japanese commercial varieties of Chinese cabbage. The present study showed that multiple copies of nonfunctional *PUI1* were located within and in the vicinity of the *UI* locus region, and that the functional *SUI1* was highly conserved in Chinese cabbage. In addition, we found a novel nonfunctional *SUI1* allele with a dominant negative effect on the functional *SUI1* allele in the heterozygote.

## 1. Introduction

Most Japanese cultivars of Chinese cabbage (*Brassica rapa* L.) are F_1_ hybrids. Traditionally, their seeds have been produced using the *Brassica* self-incompatibility (SI) system. The SI system in *Brassica* is sporophytically controlled by a single *S*-locus with highly variable, multiple alleles [1]. The *S*-locus region contains two genes, *SRK* and *SP11*/*SCR*, which correspond to female and male *S* determinants, respectively [2]. *SRK* encodes a transmembrane receptor kinase, which is expressed specifically in stigma, and *SP11*/*SCR* encodes a small cysteine-rich ligand for SRK, which is localized on the pollen coat [3,4,5]. The *S*-haplotype-specific interaction of SP11 and the extracellular domain of SRK induces the SI reaction, in which the self-pollen fails to germinate or penetrate into the stigma [6]. The number of *S*-haplotypes has been estimated to be more than 100 in *B. rapa* [7,8,9]. Advanced understanding of the *S*-haplotype diversity, including dominance relationships between the haplotypes [10], is important for the efficient production of high-quality F_1_-hybrid seed in *Brassica* crops.

In addition to SI, we reported an interesting incompatibility relationship between Turkish and Japanese populations of *B. rapa* [11,12]. Pollen of the Turkish line was rejected on the stigma of the Japanese line, although crossing in the reverse direction showed compatibility. This cross-incompatibility operating in one direction, unilateral incompatibility (UI) occurred within species, in contrast to the UI that is known to occur in interspecies crossing [13,14]. Our molecular genetic studies of intraspecies UI in *B. rapa* revealed that it was controlled by the stigma-expressed gene, *stigmatic unilateral incompatibility 1*, *SUI1*, encoding an SRK-like receptor kinase and the pollen-expressed gene, *pollen unilateral incompatibility 1*, *PUI1*, encoding an SP11-like small cysteine-rich ligand. *SUI1* and *PUI1* are tightly linked and are considered to originate from a duplication event of the *SRK*-*SP11* region in *Brassica* [12]. The *S* locus is located on chromosome A07, while the *UI* locus (containing *SUI1* and *PUI1*) is on chromosome A04 of *B. rapa* [12]. From our further analysis of genetic diversity and distribution of the *PUI1* and *SUI1* genes in *B. rapa*, a functional *PUI1-1* allele was found only in the Turkish lines and not in the Japanese lines, while the three functional *SUI1* alleles (*SUI1-1*, *-2,* and *-3*) were found in Japanese wild populations and some cultivated varieties. Thus, loss of function of *SUI1* in Turkish lines and *PUI1* in Japanese lines might have resulted in the unidirectional pollen-stigma incompatibility in *B. rapa* [12].

The physiological pollen-rejection phenotype of the intraspecies UI is similar to that of SI and is consistent with the involvement of *M*-locus protein kinase (MLPK) in UI, which may function in SRK-mediated SI signal transduction [15,16]. It is noteworthy that the incompatibility response of UI is almost as strong as in the rigid SI phenotype in *B. rapa*. Thus, UI may have an effect on the SI-dependent breeding process in *B. rapa*. In this study, we extensively analyzed *SUI1* and *PUI1* alleles in Japanese cultivated lines of Chinese cabbage (*Brassica rapa* var. *pekinensis*). The results presented here give new insight into the historical relationship between UI and the breeding system of Chinese cabbage in Japan.

## 2. Results

### 2.1. Cultivars of Chinese Cabbage Produced by Japanese Seed Companies

The UI phenotype observed on the stigma (stigma-side UI phenotype) was originally identified in the Japanese commercial hybrid variety ‘Osome’ of Japanese mustard spinach, Komatsuna (*B. rapa* var. *perviridis*), from the Takii seed company [11]. To understand the role of *SUI1* in Japanese *B. rapa* cultivars, here we examined 52 commercial cultivars of Chinese cabbage (*B. rapa* var. *pekinensis*) from 16 Japanese seed companies (listed in Table 1) to determine their *SUI1* and *PUI1* alleles in addition to their stigma-side UI phenotype. All the cultivars used in this study, except ‘Kashinhakusai’ (#8), are F_1_ hybrids. Because functional *SUI1* alleles behave as dominant over nonfunctional alleles [11], they can be analyzed to predict the UI phenotype on the stigma side of hybrid varieties.

### 2.2. Stigma-Side UI Phenotype Determined by Pollination Test

To verify the stigma-side UI phenotype of the Japanese cultivars of Chinese cabbage, stigmas of 47 cultivars were crossed with the pollen from the Turkish line (*S^24^t*, *S^40^t,* or *S^21^t*) possessing *PUI1-1*/*PUI1-1* with crossing combinations of different *S*-haplotypes for discriminating the UI phenotype from the SI phenotype (*S^21^t* was produced for this study) [16]. Among the 47 cultivars, 85% (40 cultivars) had the incompatibility (UI) phenotype to the Turkish pollen (Table 1). Only seven cultivars, ‘Chihiri 70′ (#5), ‘Eiki’ (#14), ‘Kasumihakusai’ (#17), ‘Gokui’ (#45), ‘Hakuei hakusai’ (#50), ‘Taibyou apolo 60′ (#83), and ‘Super CR Shinrisou’ (#88), had the compatibility (UC) phenotype to the Turkish pollen (Table 1). Thus, the majority of the Chinese cabbage cultivars we tested have the ability to reject the *PUI1-1*/*PUI1-1* pollen, indicating that they possess functional *SUI1* allele(s).

### 2.3. The SUI1 Allele and Its Distribution

We isolated the full-length *SUI1* gene by polymerase chain reaction (PCR) amplification from the genomic DNA of each cultivar and determined its allele(s) by sequencing, as listed in Table 1. From 22 cultivars in which *SUI1* was sequenced, six alleles, including functional alleles (*SUI1-1* and *-2*), were identified. One cultivar, ‘Kashinhakusai’ (#8), with stigmatic UI phenotype, had the *SUI1-1* allele, which was originally isolated from Komatsuna variety ‘Osome’ [11,12]. This may be because, among the cultivars used in the present study, only ‘Kashinhakusai’ (#8) is not an F_1_ hybrid, as described above. The 16 cultivars with stigmatic UI phenotype possessed the *SUI1-2* allele (Table 1), which has been found in wild *B. rapa* populations [11,12]. Three alleles encoding putative intact SUI1 proteins, *SUI1-10* (accession, LC641787)*, SUI1-11* (accession, LC641786), and *SUI1-12* (accession, LC641785)*,* and one allele encoding truncated protein, *sui1-t10* (accession, LC641784) were newly identified alleles in this study (Figure 1). Phylogenetic analysis with amino acid sequences revealed that *SUI1-11* and *SUI1-12* belonged to the same clade, and this was different from the clade with the functional SUI1s (SUI1-1, -2, and -3) and SUI1-10 (Figure 2), suggesting that *SUI1-11* and *SUI1-12* are nonfunctional alleles. Four out of seven stigmatic UC cultivars possessed *SUI1-10*, *-12*, or *sui1-t10*.

In the case of the *SUI1-10* allele, found in cultivars ‘Hakuei hakusai’ (#50) and ‘Taibyou apolo 60′ (#83), a single base substitution at codon 413 (changing the residue from cysteine to tyrosine) was present at the C-terminus of the extracellular domain (Figure 1). Both cultivars possessing the *SUI1-10* allele showed stigmatic UC phenotype, despite being heterozygous for the functional *SUI1-2* allele (Table 1), indicating that there was a dominant negative effect of *SUI1-10* toward *SUI1-2* (as described below in detail).

On the other hand, the *SUI1-11* allele had 17 amino acid changes in the extracellular domain, and cultivars ‘Mainoumi’ (#1) and ‘Menkoi’ (#9) with *SUI1-2/SUI1-11* heterozygote showed the stigmatic UI phenotype (Figure 1, Table 1). Even if *SUI1-11* is nonfunctional, as expected, the stigmatic UI phenotype is consistent with the dominance of *SUI1-2* over *SUI1-11*.

The *SUI1-12* allele had six amino acid changes in the extracellular domain. The cultivar ‘Kasumihakusai’ (#17) had *SUI1-2* and *SUI1-12* alleles as a heterozygote, and it showed the stigmatic UC phenotype (Figure 1, Table 1). It is also possible that *SUI1-12* might show a dominant negative effect to *SUI1-2* in ‘Kasumihakusai’ (#17), as in the case of *SUI1-10* in ‘Hakuei hakusai’ (#50) and ‘Taibyou apolo 60′ (#83).

‘Chihiri 70′ (#5) possessed the truncated *sui1-t10* allele (Figure 1). All the 15 *SUI1* clones of ‘Chihiri 70′ (#5) isolated from two independent PCR amplifications were *sui1-t10*, suggesting that ‘Chihiri 70′ (#5) is homozygous for *sui1-t10*, which is consistent with its stigmatic UC phenotype. The sequence of the extracellular domain of *sui1-t10* was perfectly matched with *SUI1-1* and *SUI1-2* functional alleles, but there was a 10-bp deletion in the sixth exon, as in *sui1-t4*, *sui1-t5,* and *sui1-t6*, which results in a frameshift and creates a premature termination codon [12].

### 2.4. The PUI1 Allele and Its Distribution

To examine the *PUI1* alleles of 48 cultivars of Chinese cabbage, we cloned the PCR fragments of the full-length *PUI1* and determined their sequences (Table 1, see Materials and Methods section). In these Japanese cultivars, we found three nonfunctional alleles (*pui1-3*, *-4,* and *-6*), which have been reported previously [12]. Out of the 48 cultivars, 34 possessed both *pui1-3* and *pui1-4*, and 14 possessed all three alleles (Table 1). The existence of three alleles in an individual plant indicates the possibility of duplication of *PUI1*. To verify this duplication, we first propagated the self-pollinated progeny of ‘Super CR Shinrisou’ (#88; *pui1-3*/*pui1-**4*) and determined the *PUI1* genotype of the 22 segregants using a direct sequencing method. It was found that all the segregants exhibited both *pui1-3* and *pui1-4*, suggesting that the *pui1-3* and *pui1-4* genes were linked and homozygous in this progeny (Table 2). Next, we propagated the self-pollinated progeny of ‘Gokui’ (#45; *pui1-3*/*pui1-**4/pui1-6*) and determined the *PUI1* genotype of the 32 segregants using a PCR-restriction fragment length polymorphism (RFLP) method. It was found that all individuals possessed *pui1-3*, *pui1-4*, and *pui1-6*, suggesting that the three *PUI1* genes (*pui1-3*/*pui1-**4/pui1-6*) were linked and homozygous in this progeny (Table 2). Furthermore, a similar PCR-RFLP experiment was performed using ‘Nanzan’ (#101; *pui1-3*/*pui1-4/pui1-6*) selfed progeny (Table 2, Appendix A). Interestingly, the self-pollinated population (78 plants) of ‘Nanzan’ segregated to *pui1-3*/*pui1-**4* and *pui1-3*/*pui1-**4/pui1-6* plants. Their segregation ratio was 17:61 (1:3; chi-square test, χ^2^ = 0.43, *p* > 0.05) fit for a simple Mendelian inheritance. The result indicates that ‘Nanzan’ (#101) is a heterozygote of duplicated (*pui1-3*/*pui1-**4*) and triplicated (*pui1-3*/*pui1-4/pui1-6*) *PUI1* genes in the *B. rapa* genome. Thus, duplication and/or triplication of nonfunctional *PUI1* genes had occurred at the *UI* locus region in Japanese *B. rapa* cultivars.

### 2.5. Genetic Segregation Analysis of the Dominant Negative Effect of SUI1-10

As described above, ‘Hakuei hakusai’ (#50) and ‘Taibyou apolo 60′ (#83), possessing the *SUI1-2*/*SUI1-10* genotype, exhibited stigma-side UC phenotype (i.e., accepting the Turkish *PUI1-1/PUI1-1* pollen), even though they have a functional *SUI1-2* allele. To confirm this dominant negative effect of *SUI1-10*, we performed a genetic analysis of ‘Taibyou apolo 60′ (#83).

We produced self-pollinated progeny of ‘Taibyou apolo 60′ (#83-S_1_ progeny) and determined their stigma-side UI phenotype and *SUI1* genotype (Table 3). Stigma-side UI phenotypes of this progeny were determined by test cross-pollination using *PUI1-1*/*PUI1-1* homozygous plants (*S*^24^*t*) as the pollen donor. The *SUI1-2* and *SUI1-10* alleles were discriminated by direct-sequencing detection of a single nucleotide polymorphism at codon 413 and were segregated in the #83-S_1_ progeny; three of eleven plants showed stigma-side UI, and the others were stigma-side UC. Stigma of thee *SUI1-2*/*SUI1-2* homozygous plants were incompatible to the *S*^24^*t* pollen (UI), and five *SUI1-2*/*SUI1-10* heterozygous, and three *SUI1-10*/*SUI1-10* homozygous individuals showed compatible pollen tube penetration with the *S*^24^*t* pollen (UC), indicating that *SUI1-10* is nonfunctional and has a dominant negative effect to the functional *SUI1-2*.

For further confirmation of this effect, the #83-S_2_ population with a higher number of plants was produced by self-bud pollination of the #83-S_1_ *SUI1-2/SUI1-10* heterozygous plants. In the #83-S_2_ population, *SUI1* genotypes segregated as expected; for genotypes *SUI1-2/SUI1-2*: *SUI1-2/SUI1-10*: *SUI1-10/SUI1-10* the observed ratio was 23:41:16 (1:2:1; chi-square test, χ^2^ = 1.27, *p* > 0.05, *df* = 2, Table 3, Appendix A). The stigma-side UC phenotype and *SUI1-10* genotype of the #83-S_2_ population showed perfect linkage in the 80 plants (Table 3). Thus, it was concluded that the nonfunctional *SUI1-10* does show a dominant negative effect on the functional *SUI1-2*.

To verify if this effect is observed with the other functional allele, we produced *SUI1-3/SUI1-10* heterozygous plants by a cross between *SUI1-3/SUI1-3* [11,12] and *SUI1-10/SUI1-10* plants selected from the #83-S_2_ population. Stigmas of *SUI1-3/SUI1-10* heterozygous plants were compatible (UC) with *PUI1-1/PUI1-1* pollen from the *S*^24^*t* and also *S*^40^*t* lines, indicating that *SUI1-10* also shows a dominant negative effect on the functional *SUI1-3*.

## 3. Discussion

Highly controlled pollen-stigma incompatibility is important for F_1_ hybrid seed production of *Brassica* cultivars. The molecular mechanism of SI in *Brassica* has been studied for many years and is used in F_1_ breeding. The recently discovered UI system, regulated by *SUI1* and *PUI1,* can potentially provide another mechanism to control pollination in *B. rapa*. Therefore, determination of the *UI* genotype is considered as important as the SI genotype in the breeding of this major Japanese vegetable, Chinese cabbage. In this study, we determined the *SUI1* and *PUI1* allelic diversity of 22 and 48 cultivars, respectively, of Chinese cabbage in Japan. In addition, we confirmed the stigma-side UI phenotype of 47 cultivars. This revealed that most of the cultivars showed a stigma-side UI phenotype with a functional *SUI1* allele (*SUI1-2*), whereas no functional *PUI1* allele (*PUI1-1*) was found. We also searched the re-sequence data of *B. rapa* lines that are stocked at Chungnam National University and found a functional *SUI1-2* allele in a South Korean population (data not shown). The fact that functional *SUI1* alleles are present in Japanese and South Korean cultivars should be taken into consideration in breeding programs for *B. rapa*. UI may be beneficial as the additional incompatibility, which could be used in breeding programs by the introduction of *PUI1-1* to the pollen donor.

To the best of our knowledge, there is no report that traits important for Chinese cabbage are mapped to flanking regions of the *UI* locus in chromosome A04. Thus, for an unknown reason, the functional *SUI1-2* has been selected, and its sequence has been conserved during the breeding of Chinese cabbage cultivars in Japan. It would be interesting to investigate whether *SUI1* itself strengthens SI and thus increases the efficiency of F_1_ seed production.

In our previous study, we isolated nine intact alleles of *SUI1* and showed that *SUI1-1*, *SUI1-2*, and *SUI1-3* are incompatible with *PUI1-1*/*PUI1-1* pollen [12]. *SUI1-1* was originally isolated from a Japanese commercial hybrid variety of Komatsuna (*B. rapa* var. *perviridis*), and *SUI1-2* and *SUI1-3* were found in Japanese wild populations of *B. rapa* [12]. In the current study, we isolated three novel intact *SUI1* alleles; one (*SUI1-10*) belongs to the functional clade (with *SUI1-1*, *SUI1-2*, and *SUI1-3*) and the other two alleles (*SUI1-11* and *SUI1-12*) belong to the nonfunctional clade (Figure 2). The fact that *SUI1-10*/*SUI1-10* homozygote is stigmatic UC indicates that *SUI1-10* is a nonfunctional allele (Table 3). The Cys-413 residue of *SUI1-2* is the last of the 12 highly conserved cysteine residues in the *SUI1* extracellular domain and is located within the PAN_APPLE domain, which is the C terminal region of the extracellular receptor region. It has been clarified that homodimerization of SRK in Brassicaceae is essential for ligand interaction [17]. The PAN_APPLE domain of SRK has been shown to be important for ligand-independent dimer formation of SRKs and is responsible for correct intracellular trafficking [18,19,20,21]. It has been reported that the last Cys residue of SRK is predicted to form an intramolecular disulfide bond [20,21]. Thus, although the *SUI1-10* sequence is similar to the functional *SUI1-2*, the C413Y mutation of *SUI1-10* might cause structural disruption of *SUI1* and breakdown of incompatibility through unusual dimer formation.

A feature of the sporophytic regulation of SI is the dominance relationship between *S*-haplotypes [10,22,23]. The molecular mechanism of the pollen-side dominance relationship has been well studied and revealed that mono-allelic gene expression of the dominant *SP11* haplotype is controlled by small RNA-based epigenetic regulation [24,25,26]. On the stigma side, there is a complex allelic interaction that is as yet unexplained [10]. It was presumed that the SRK protein itself determines the dominance relationship rather than differences in *SRK* gene expression [23], and Naithani et al. [18] noted that the stigma-side dominance relationship may result from an increased tendency for heterodimer formation in some SRK pairs [18]. On the other hand, the existence of dominant negative alleles of receptor kinases that function as receptor complexes in many situations during plant development is widely known [27,28,29]. In most of these, the formation of a receptor complex with abnormal receptor proteins or receptor-related proteins encoded by dominant negative alleles causes disruption of signaling pathways. Thus, one possible explanation for the dominant negative effect of *SUI1-10* may be an increase of *SUI1-2*/*SUI1-10* heterodimer on the stigma surface and competitive inhibition of the interaction with the *PUI1* ligand. We also found a dominant negative effect of *SUI1-10* to *SUI1-3*, which has four aa substitutions (R322H, I326L, R363H, and V364D) compared to the extracellular domain of *SUI1-2*, suggesting that these four residues are not important for the effect.

In this study, it was found that the *PUI1* gene of Japanese cultivars of Chinese cabbage showed very low diversity. Among six *PUI1* alleles, of which only *PUI1-1* from a Turkish strain can induce UI [12], only two patterns of genotype (*pui1-3/pui1-4* or *pui1-3/pui1-4/pui1-6*) were observed, and no cultivars with a functional *PUI1-1* allele could be found. Interestingly, the *pui1-3/pui1-4* genotype might consist of two linked *pui1-3* and *pui1-4* genes (Appendix A). Similarly, the *pui1-3/pui1-4/pui1-6* genotype might consist of three linked *pui1-3, pui1-4,* and *pui1-6* genes (Appendix A). Such duplication and triplication of nonfunctional *PUI1* have complicated the *UI* locus region. Although such *PUI1* duplication or triplication cannot be found in the reference genome information of *B. rapa* inbred line Chiifu (*B. rapa* reference genome version 3.0, https://brassicadb.cn, accessed on 1 April 2021), de novo genomic sequence assembly of these Chinese cabbage cultivars using next-generation sequencing technology, including long-read sequencing, would provide new insights into the genomic structure of the *UI* locus [30]. In fact, we can find the two duplicated *PUI1* genes on the *UI* locus of the genome sequence of *B. rapa* Z1(version 1.0, https://brassicadb.cn, accessed on 19 October 2021, Appendix A) [31].

Further analysis of the genetic diversity of the *UI* locus in *B. rapa* other than Chinese cabbage (subsp. *pekinensis*), such as turnips (subsp. *rapa*), leafy *Brassica* crops (subsp. *chinensis*, *periridis*), and field mustard (subsp. *oleifera*) will not only contribute to the discovery of novel alleles but also provide new insights into the genomic structure of the pollen-side factor and the dominant recessive interaction of the stigma-side factor. It will also be interesting to determine whether the *UI* locus has a multi-allelic structure like the *S* locus.

## 4. Materials and Methods

### 4.1. Plant Material

The plant material consisted of 52 commercial cultivars of Chinese cabbage, *B. rapa* ssp. *pekinensis* (Table 1). All except one, ‘Kashinhakusai,’ were F_1_ hybrid cultivars. To produce self-pollinated progeny, bud pollination was performed. Petals and stamens were removed from a young flower bud (2–4 d before flowering), and the immature pistil was pollinated. The pollinated pistil was then covered with a paper bag until the seed was harvested. Plant materials were vernalized at 4 ^o^C for 4 weeks in a refrigerator and then grown in a greenhouse.

### 4.2. Test Pollination

Flower buds were cut at the peduncle and pollinated. After pollination, they were stood on 1% solid agar for about 24 h under room conditions. Then, pistils of the pollinated flowers were softened in 1N NaOH for 1 h at 60 °C and stained with basic aniline blue (0.1 M K_3_PO_4_, 0.1% aniline blue). Samples were mounted in 50% glycerol on slides and observed by UV fluorescence microscopy (Appendix A) [32]. At least three flowers were used from each cross combination, and observations were generally replicated at least three times on different dates for each cross combination. For the determination of the stigma-side UI phenotype, *PUI1-1*/*PUI1-1* homozygous plants (*S^24^t*, *S^40^t, and S^21^t*) were used as the pollen donor in test pollinations (*S^21^t* was produced for this study) [16].

### 4.3. Cloning, Sequencing, and Genotyping of SUI1and PUI1 Alleles

Total DNA was extracted from young leaf tissue of *B. rapa* by the procedure of Murray and Thompson (1980) or using a DNeasy plant mini kit (Qiagen) [33]. For molecular cloning of full-length *SUI1* and *PUI1* genes, genomic PCR was performed using KOD-Plus-Neo DNA polymerase (TOYOBO) according to the manufacturer’s instructions. PCR primers SUI1cDNA_F3 and SUI1_gR2 for *SUI1* and PCP-like1-F1 and PCP-like1-R1 for *PUI1* were used (Appendix A). All amplified fragments were detected as a single band in the gel electrophoresis. PCR products were modified by adding 3′-A overhangs using A-attachment mix (TOYOBO) and cloned into a vector, pTAC-2, using DynaExpress TA PCR Cloning kit (Biodynamics). The nucleotide sequence was determined with a 3500 or 310 Genetic Analyzer using Big Dye Terminator version 3.1 or 1.1 Cycle Sequencing Kit (Applied Biosystems); in the case of *SUI1*, the *SUI1*-specific sequencing primers, SUIcDNA_F3, SUI_gR2, SUIinter_cF1, SUIinter_cF2, SUIinter_cF3, SUIinter_GF1, SUI1inter_cF4, and SUIinter_cF5 (Figure 1 and Appendix A), were used. GENETYX version 13 software package (GENETYX Corp.) was used for the sequence comparison and alignment. For the segregation analysis, we determined the genotype of *SUI1* and *PUI1* alleles by direct sequencing of PCR products. *SUI1-1* and *SUI1-10* alleles were amplified using primers SUI1_2-10typeSDF and SUI1_2-10typeSDR (Appendix A). Each *PUI1* allele was amplified using the primer pair for the *PUI1* second exon region, PUI1-3.4.6-F, and PUI1-3.4.6-R (Appendix A). For discrimination of *PUI1* alleles by PCR-RFLP, amplified DNA fragments were cut by restriction enzyme (BamHI, SalI, or BsrI), followed by checking on an electrophoresed agarose gel (Appendix A). For the direct sequencing marker, amplified fragments were purified from the electrophoresed agarose gel and sequenced as described above.

### 4.4. Phylogenetic Analysis

Phylogenetic analysis was performed on the Phylogeny.fr platform (http://www.phylogeny.fr/, accessed on 21 October 2021) [34]. Full-length amino acid sequences were aligned with MUSCLE (version 3.7) configured for the highest accuracy. Accession numbers of SRKs and SUI1s are listed in Appendix A. After alignment, ambiguous regions were removed with Gblocks (version 0.91b). The phylogenetic tree was reconstructed using the PhyML program (version 3.0 aLRT). The default substitution model was selected assuming an estimated proportion of invariant sites and 4 gamma-distributed rate categories to account for rate heterogeneity across sites. The reliability of internal branches was assessed using the bootstrapping method (100 bootstrap replicates). The tree was represented with TreeDyn (version 198.3).

## Figures and Tables

**Figure 1 plants-10-02467-f001:**
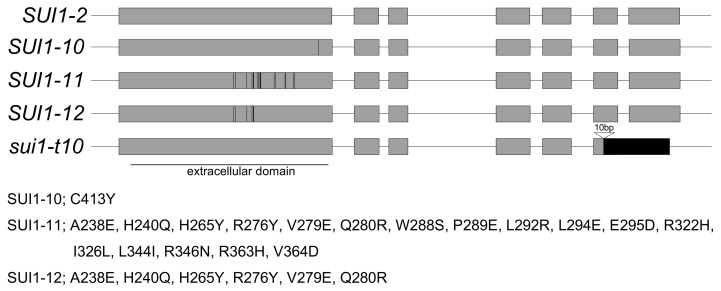
Schematic representation of the *SUI1* genomic sequences in this study. The shaded boxes represent the protein coding regions. Positions of amino acid substitutions compared to *SUI1-2* are shown by bars and listed below. The extracellular domain (consisting of most of the 1st exon) is indicated. The position of the 10-bp deletion of *sui1-t10* is shown in the sixth exon.

**Figure 2 plants-10-02467-f002:**
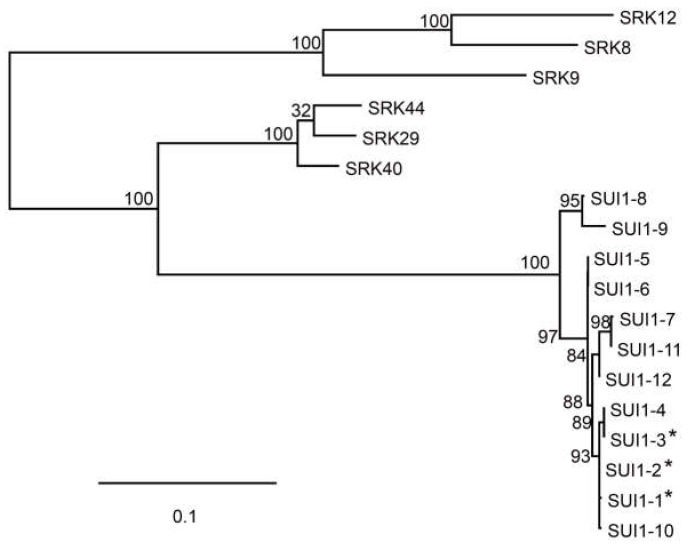
A maximum likelihood phylogenetic tree of SUI1s and SRKs in *B. rapa*. Branch support values from 100 bootstraps are indicated. Functional SUI1s that genetically interact with PUI1-1 are indicated by asterisks (*).

**Table 1 plants-10-02467-t001:** UI phenotype and genotype of Japanese cultivars of Chinese cabbage.

Sample Number	Seed Company	Cultivar	Stigma-Side UI Phenotype	Genotype	
				*SUI1*	*PUI1*
#1	Tokita Seed Co., Ltd.	Mainoumi	UI	*SUI1-2/SUI1-11*	*pui1-3/pui1-4/pui1-6*
#2	Takii & Co., Ltd.	Puchihiri	UI	*SUI1-2*	*pui1-3/pui1-4*
#3	Takii & Co., Ltd.	Kigokoro 75	UI	*SUI1-2*	*pui1-3/pui1-4*
#4	Sakata Seed Corp.	Kimikomachi	UI	*SUI1-2*	*pui1-3/pui1-4*
#5	Takii & Co., Ltd.	Chihiri 70	UC	*sui1-t10*	*pui1-3/pui1-4*
#6	Takii & Co., Ltd.	Banki	UI	*SUI1-2*	nd
#7	Watanabe Seed Co., Ltd.	Matsushima shin2gou	UI	nd	*pui1-3/pui1-4*
#8	Noguchi Seed Co.	Kashinhakusai	UI	*SUI1-1*	*pui1-3/pui1-4*
#9	Watanabe Seed Co., Ltd.	Menkoi	UI	*SUI1-2/SUI1-11*	*pui1-3/pui1-4/pui1-6*
#10	Ishii Seed Growers Co., Ltd.	Kinami 90	UI	nd	nd
#11	Kaneko Seed Co., Ltd.	Kougetsu 77	UI	*SUI1-2*	*pui1-3/pui1-4*
#14	Kaneko Seed Co., Ltd.	Eiki	UC	nd	*pui1-3/pui1-4/pui1-6*
#16	Kaneko Seed Co., Ltd.	Moeki	UI	nd	*pui1-3/pui1-4*
#17	Kaneko Seed Co., Ltd.	Kasumihakusai	UC	*SUI1-2/SUI1-12*	*pui1-3/pui1-4/pui1-6*
#18	Kaneko Seed Co., Ltd.	Shouki	UI	nd	*pui1-3/pui1-4*
#19	Watanabe Seed Co., Ltd.	Strong CR	UI	*SUI1-2*	*pui1-3/pui1-4*
#20	Watanabe Seed Co., Ltd.	Aiki	UI	nd	*pui1-3/pui1-4/pui1-6*
#23	Nozaki Saishujo Ltd.	Maiko	nd	nd	*pui1-3/pui1-4*
#24	Nozaki Saishujo Ltd.	Chi China	UI	nd	nd
#25	Nozaki Saishujo Ltd.	Eisyun	nd	nd	*pui1-3/pui1-4/pui1-6*
#27	Nozaki Saishujo Ltd.	Retasai	UI	nd	*pui1-3/pui1-4/pui1-6*
#33	Marutane Seed Co., Ltd.	Chikara	UI	*SUI1-2*	*pui1-3/pui1-4*
#35	Yamato Noen Co., Ltd.	Kiyorokobi	UI	*SUI1-2*	*pui1-3/pui1-4*
#41	Watanabe Seed Co., Ltd.	Kiai 65	UI	nd	*pui1-3/pui1-4*
#45	Kaneko Seed Co., Ltd.	Gokui	UC	nd	*pui1-3/pui1-4/pui1-6*
#47	Kaneko Seed Co., Ltd.	Taibyou nozomi 60	UI	nd	*pui1-3/pui1-4*
#49	Mikado Kyowa Seed Co., Ltd.	CR Ouken	UI	*SUI1-2*	*pui1-3/pui1-4*
#50	Mikado Kyowa Co., Ltd.	Hakuei hakusai	UC	*SUI1-2/SUI1-10*	*pui1-3/pui1-4*
#51	Mikado Kyowa Co., Ltd.	Senki	nd	nd	*pui1-3/pui1-4/pui1-6*
#53	Sakata Seed Corp.	Saiki	nd	nd	*pui1-3/pui1-4/pui1-6*
#55	Sakata Seed Corp.	Yumebuki	UI	nd	*pui1-3/pui1-4/pui1-6*
#57	Takayama Seed Co., Ltd.	Gokigen	UI	nd	*pui1-3/pui1-4*
#58	Ishii Seed Growers Co., Ltd.	CR Seiga 65	UI	*SUI1-2*	*pui1-3/pui1-4*
#62	Takii & Co., Ltd.	Oushou	UI	nd	nd
#63	Takii & Co., Ltd.	Musou	UI	*SUI1-2*	*pui1-3/pui1-4*
#64	Takii & Co., Ltd.	Senshou	UI	nd	*pui1-3/pui1-4*
#65	Takii & Co., Ltd.	Kinshou	UI	nd	*pui1-3/pui1-4*
#74	Tohoku Seed Co., Ltd.	Daifuku	UI	*SUI1-2*	*pui1-3/pui1-4*
#75	Tohoku Seed Co., Ltd.	Daifuku75	UI	nd	*pui1-3/pui1-4*
#77	Tohoku Seed Co., Ltd.	Shinseiki	UI	nd	*pui1-3/pui1-4*
#80	Nanto Seed Co., Ltd.	CR Kinshachi 75	UI	*SUI1-2*	*pui1-3/pui1-4*
#83	Nanto Seed Co., Ltd.	Taibyou apolo 60	UC	*SUI1-2/SUI1-10*	*pui1-3/pui1-4*
#84	Nippon Norin Seed Co.	Kikumusume 65	UI	*SUI1-2*	*pui1-3/pui1-4*
#85	Nippon Norin Seed Co.	Kien75	UI	nd	*pui1-3/pui1-4*
#88	Nippon Norin Seed Co.	Super CR Shinrisou	UC	*SUI1-2*	*pui1-3/pui1-4*
#91	Takayama Seed Co., Ltd.	Kinkaku 65	UI	nd	*pui1-3/pui1-4/pui1-6*
#96	Tokita Seed Co., Ltd.	Haruhi	UI	nd	*pui1-3/pui1-4/pui1-6*
#97	Nanto Seed Co., Ltd.	Taiki 60	UI	nd	*pui1-3/pui1-4*
#101	Musashino Seed Co., Ltd.	Nanzan	UI	nd	*pui1-3/pui1-4/pui1-6*
#102	Watanabe Seed Co., Ltd.	Seitoku	nd	nd	*pui1-3/pui1-4*
#103	Watanabe Seed Co., Ltd.	Shunjuu	UI	nd	*pui1-3/pui1-4*
#104	Watanabe Seed Co., Ltd.	Kaname	UI	nd	*pui1-3/pui1-4*

nd, not determined.

**Table 2 plants-10-02467-t002:** Segregation analysis of *PUI1* allele in the selfed progeny of #45, #88, and #101.

Sample Number	Cultivar	*n*	*PUI1* Genotype Detected	
			*pui1-3/pui1-4*	*pui1-3/pui1-4/pui1-6*
#45	Gokui	32	0	32
#88	Super CR Shinrisou	22	22	-*
#101	Nanzan	78	17	61

*, ‘Super CR Shinrisou’ (#88) does not have *pui1-6* allele.

**Table 3 plants-10-02467-t003:** Segregation analysis of *SUI1* allele in the selfed progeny of #83.

Population	*SUI1* Genotype	*n*	Stigma-Side UI Phenotype	
			UI	UC
#83-S_1_	*SUI1-2/SUI1-2*	3	3	0
	*SUI1-2/SUI1-10*	5	0	5
	*SUI1-10/SUI1-10*	3	0	3
#83-S_2_	*SUI1-2/SUI1-2*	23	23	0
	*SUI1-2/SUI1-10*	41	0	41
	*SUI1-10/SUI1-10*	16	0	16

## Data Availability

The available data are presented in the manuscript.

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
