# Peer review of "Genetic Diversity of Genes Controlling Unilateral Incompatibility in Japanese Cultivars of Chinese Cabbage"

_plants, 2021, doi:10.3390/plants10112467_

Round 1

Reviewer 1 Report

The manuscript “Genetic Diversity of Genes Controlling Unilateral Incompatibility in Japanese Cultivars of Chinese Cabbage” reports the phenotypic and genetic analysis of a group of more than 40 cultivars of Chinese cabbage for unilateral incompatibility and alleles at SUI1/PUI1 loci.

There are some aspects to be improved.

Introduction, please give more details on chromosome location UI locus.

The authors state that the physical organization of the locus it is not known in terms of order and structure (physical distances). They should check the available genome sequences for Chinese cabbage and related species to investigate this locus and report in the Results and Discussion sections, even if conclusive information is not retrieved.

Another point is that it is not clear how the PUI1 sequences were obtained: the same primer couple was used to amplify the three forms? Were they characterized by different size and separated by gel electrophoresis, or a single band was obtained and cloned? All these details must be added in Results. In Fig. S1 add gene length, and intron/exon structure.

UV fluorescence microscopy was used to assess incompatibility. Add some pictures showing examples of compatibility and incompatibility.

Reviewer 2 Report

This manuscript describes and investigation into the genetic diversity of the SUI1 and PUI1 genes the control unilateral incompatibility in Japanese B. rapa, and investigated the UI phenotype associated with these alleles.

This is an excellently crafted set of experiments and carefully prepared manuscript. The results are interesting and of agricultural application in the generation of F1 hybrid seed. Of particular interest is the identification of a novel nonfunctional SUI1 allele with a dominant negative effect on the functional SUI1 allele in heterozygotes, that could be useful for hybrid seed production.

I thank the authors for the care taken in preparing the manuscript - it was a joy to read. I did find a few minor typographical/spelling errors, which the authors should correct prior to publication:

Line 33 “seeds have been produced traditionally using” should be changed to “traditionally seeds have been produced using”

Line 246 “dominant relationships” should be “dominance relationships”

Line 252 “Naithani et al. (2007) noted” should be “Naithani et al. [18] noted”

Line 282 “the SI locus.” should be “the S-locus.”

Line 427 “ Chory, A.” should be “Chory, J.”

Round 2

Reviewer 1 Report

The ms has been revised as requested.